# Leadership development among public health officials in Nepal: A grounded theory

**Sudarshan Subedi** [1,2]*, **Colin MacDougall**[2], **Darlene McNaughton**[3], **Udoy Saikia**[4], **Tara Brabazon**[5]

**1** School of Health and Allied Sciences, Pokhara University, Pokhara, Nepal, **2** College of Medicine and Public Health, Flinders University, Adelaide, South Australia, **3** School of Humanities, Arts, and Social Sciences, University of New England, Armidale, New South Wales, Australia, **4** College of Humanities, Arts and Social Sciences, Flinders University, Adelaide, South Australia, **5** Office of Graduate Research, Flinders University, Adelaide, South Australia

* subedisudarshan@gmail.com

**Data Availability Statement:** All relevant data (interview guides, sample of memo writing, examples of codes) are within the manuscript and its Supporting Information files. Any personal

## Abstract

Leadership in public health is necessary, relevant, and important as it enables the engagement, management, and transformation of complex public health challenges at a national level, as well as collaborating with internal stakeholders to address global public health threats. The research literature recommends exploring the journey of public health leaders and the factors influencing leadership development, especially in developing countries. Thus, we aimed to develop a grounded theory on individual leadership development in the Nepalese context. For this, we adopted constructivist grounded theory, and conducted 46 intensive interviews with 22 public health officials working under the Ministry of Health, Nepal. Data were analysed by adopting the principles of Charmaz's constructivist grounded theory. The theory developed from this study illustrates four phases of leadership development within an individual–initiation, identification, development, and expansion. The 'initial phase' is about an individual's wishes to be a leader without a formal role or acknowledgement, where family environment, social environment and individual characteristics play a role in influencing the actualisation of leadership behaviours. The 'identification phase' involves being identified as a public health official after having formal position in health-related organisations. The 'development' phase is about developing core leadership capabilities mostly through exposure and experiences. The 'expansion' phase describes expanding leadership capabilities and recognition mostly by continuous self-directed learning. The grounded theory provides insights into the meaning and actions of participants' professional experiences and highlighted the role of individual characteristics, family and socio-cultural environment, and workplace settings in the development of leadership capabilities. It has implications for academia to fulfill the absence of leadership theory in public health and is significant to fulfill the need of leadership models grounded in the local context of Asian countries.

identifiable information has been excluded from the extracts to ensure participants' anonymity.

**Funding:** The author(s) received no specific funding for this work. However, the corresponding author (SS) has received a scholarship from Flinders University for his doctorate degree and this study is a part of that degree.

**Competing interests:** The authors have declared that no competing interests exist.

# Background

Leadership is one of the most observed phenomena but least understood [1]. Despite numerous research and studies, leadership lacks universal definition because of its personalized application and different ways to influence people. To understand it simply, leadership is 'a process whereby an individual influence a group of individuals to achieve a common goal' [2]. It is also defined as 'the influencing process between leaders and followers to achieve organisational objectives through change' [3]. The review of literature revealed leadership definitions focusing on various dimensions such as centralization and/or control [4,5], interpersonal and/or social influence [2,3,6–12], relationship with others [13,14], changing behaviour of others [15], transforming others [1], mobilizing others [1,16], meaningful direction [17–19], and having followers [3,20,21].

Leadership is complex. So is public health. Public health reaches beyond health issues/problems or health services/system, and leadership is required to respond the social and political factors that impact the health of the population. Nations with scarce financial and human resources are struggling to improve public health and services [22]. Common issues faced by these countries relate to human resources management, financial management, governance, and leadership [23–27]. Throughout the 21st century, leadership in public health has been emphasized at the national level [28–33] to address the increasingly complex challenges [34], to defend against global public health threats [35], to increase the access to health services, and to deal with social issues such as poverty, stigma, and disasters [36]. Thus, a better understanding of public health leadership is needed to enable leaders to deal with contemporary public health issues [37], and to bring desirable changes in the organisation. However, officials handling leadership roles may not exercise leadership effectively if they concentrate on micromanagement of activities that impede their vision for creativity and spontaneity [38]. In most of the developing countries like Nepal, the senior officials in health organisations are technical staff (doctors, nurses, and allied health workers) who are in-charge of managing people and programs. These officials are not formally employed as leaders with the responsibility to bring changes in the organization. Rather they are employed as health workers to prevent disease and/or to promote health but while working, they are obliged to exercise or enact leadership. Stronger leadership is needed to accelerate improvements in public health, and this will happen when the practitioners of public health have commitment and competencies to perform leadership [39]. Thus, it is necessary to understand the perspectives of leadership among public health professionals from varied backgrounds to explore the way they lead public health services.

Despite the growing importance of leadership development, a research gap still exists in the health services [40] and studies in leadership development are still emerging [41]. Only a few studies have explored how leadership is developed and how organizational and socio-cultural factors are reflected in leadership. Our preliminary literature review found fewer leadership studies in the health sector as compared to business management and public administration. There are more studies on leadership in medicine and nursing than in public health. We found differences between healthcare (hospital/clinical settings) leadership and public health (community/non-clinical settings) leadership arising from the nature, scope and aims of each system. We conducted a narrative review by adopting systematic search on 'public health' and 'leadership' and ended up with 56 articles around the globe. Literature on public health leadership came mostly from developed countries in the form of secondary articles (comments, short communication, correspondence, reviews). There is a gap in primary research in developing countries in public health. A number of authors claim that research on leadership identity development are more centered in non-health sectors of developed countries [42–44] and

most leader development research focused on the United States [44]. This accord with a scoping review that found very little theoretical and empirical literature examining leadership in public health [45].

Faced with a lack of models and theories of public health leadership, public health professionals are applying leadership lessons generated from other disciplines [46,47]. Borrowing leadership theories and practices from other disciplines not only underestimates the special nature of public health problems but also *'the unique opportunities that makes leadership in public health a rich source of inspiration, frustration and fascination'* [46]. As leadership is context sensitive [48], leadership theory developed in one environment or circumstance may not be applicable to another because of the variations in socio-cultural environment and public health practices. Thus, it is necessary to study leadership in different settings. Studies also recommend further exploration on why and how someone becomes a leader in public health [49], how the journey of public health leaders progresses [50,51] and what might influence the leadership development and practices [51,52].

Moreover, leadership and governance are challenges in developing countries [23–26,53] and there is evidence that poor leadership affects the health system and public health of those countries [27,54,55]. In this context, exploring leadership dimensions should be an utmost priority. Based on the identified research gap and recommendations from previous studies, we aimed to develop a grounded theory on individual leadership development. The research question guided this study was *'how does someone become a leader in public health in context of Nepal?' We* explored the journey of public health officials and investigated the socio-cultural factors that support the progress of leadership development among them. The grounded theory built from this study was progressively developed without assuming a priori existing frameworks. This study is significant to understand the progress of leadership development and it fulfils the need of theory on leader and/or leadership development [56,57] as well as the need of leadership models grounded in the local context of Asian countries [44,58,59].

## Methodology and methods

The methodology for this study was adapted from Charmaz's constructivist grounded theory (2014) which assumes that 'knowledge rests on social constructions' and the studied reality is not an objective. Constructivist grounded theory is a revision of the grounded theory developed by Glaser and Strauss in 1967 [60]. It is similar to Glaser and Straus's interpretation that it adopts inductive, comparative, emergent and open-ended approach and uses earlier methodological strategies such as coding, memo-writing, and theoretical sampling. However, it differs from previous versions which are rooted in positivism and the idea of objective reality. Constructivist approach opposes the idea that research is best conducted by a neutral observer without pre-existing knowledge on the topic. Rather, it assumes that knowledge rests on social construction and *'acknowledge[s] the subjectivity and the researcher's involvement in the construction and interpretation of data'* [61]. Thus, it emphasizes on multiple realties, subjective relationship between the researcher and the participants, and the mutual construction of meaning by the researcher and the participants.

### Study settings

The study was conducted with different levels of governmental health institutions under the Ministry of Health in Nepal: local (district) level, provincial (regional) level, and federal (central) level from June 2018 to Dec 2019. The participants for this study were the public health officials who were employed in different positions of *"health inspection"* and *"public health administration"* group as categorized by the Nepal Health Services Rules and having academic

qualifications and/or experiences in public health' [62]. This criterion included the officials who had a qualification of Bachelor's/Master's degree in Public Health (despite of their educational background) and ever held a leadership position at any of the health institutions–District Health Office (DHO), District Public Health Office (DPHO), Regional Health Directorate (RHD), Department of Health Services (DHS) and Ministry of Health (MoH). This criterion excluded the health officials from hospital/clinical settings who were categorized under the group of *Medicine, Nursing, Pharmacy, Laboratory, Ayurveda, Integrated Medicine* and so forth. *(See Nepal Health Services Rules [62] for the categorization of health personnel in Nepalese health system).*

## Recruitment of participants

For the recruitment of participants, we asked the Ministry of Health to allocate a focal person. The focal person provided the list of potential participants based on the socio-demographic and job-related characteristics such as age, gender, caste/ethnicity, locality, entry level position, working institution, existing position and level, academic qualification, trainings, and work experiences. This strategy aimed to maintain confidentiality, to reduce the selection bias (since none of the authors were involved in recruitment) and to ensure information rich participants. Once the participants were identified, the focal person contacted them via email and/or direct phone calls to gauge if they were interested participating in the study. During this process, we asked the focal person not to persuade anyone to participate because of their organizational linkage and relationship. Based on the response from the participants, the focal person provided us the contact details of those who agreed to take part in our study. Then, we contacted participants for recruitment.

## Sample and sampling

We combined purposeful and theoretical sampling. We sought diversity among samples to include a range of perspectives on public health leadership, particularly considering the multicultural context of the country. When the participants were selected purposefully and interviewed, we experienced the repetition of information from $20^{th}$ interview with the $10^{th}$ participant. However, to make sure of the data saturation, we conducted ten more interviews with five participants. The initial samples selected via purposeful sampling (for data saturation) consisted of 30 interviews with 15 participants. This provided useful concepts and information on the central issue of the research. It was then followed by theoretical sampling (for theoretical saturation). The final sample comprised 46 interviews with 22 participants.

## Data collection

Participants were asked to participate in multiple in-depth interviews, the purpose of which was to explore their leadership journey and how leadership, as a capability, developed within them. An interview guide was developed based on our research question and accounting for social and cultural aspects of Nepalese society. Interviews were also informed by our document analysis of the 'job description' of public health officials and consultations with professionals having experience in Nepalese health system.

The initial interview explored the information regarding participants' childhood experiences, family environment, socio-cultural context including the role of gender and ethnicity, educational and/or professional interests and exposure in public health. Participants were also asked to share their inspiration for leadership, adoption of leadership insights, and its consequences on their field. The second interview was conducted two weeks after the first interview which focused on participants' job roles and responsibilities as well as their experiences and

understanding of successful leadership. Some questions for this interview were revised based on the previous interviews. The analysis of first and second interviews informed new questions for the final interview. The final interview explored more about key concepts and ideas towards the development of theory. The average duration of first, second and final interview was 60 minutes, 40 minutes, and 30 minutes respectively. Probing, as the essence of in-depth interviewing [63] was employed in full. The aspects of non-verbal communication like body languages, facial expressions, and voice tone were also observed, and these actions were recorded in the form of field notes. All the data were collected by the primary author. This is because of the author's locality, expertise in local language, excellent understandings with local culture, experiences with public health academia at local level as well as professional connections and in-depth knowledge of the health system of Nepal.

## Data analysis

Data analysis was started from the first day of data collection which was aided by transcription, memo writing, coding, constant comparison, and theoretical sorting, diagramming and integrating. All interviews were conducted in local language (Nepali) and transcribed as soon as they were completed. The primary author transcribed all the recordings in Microsoft Word. The transcripts were then reviewed and sent to participants via email to revise. This was done to ensure that the transcript reflects the participant's views. Each transcript was kept in its original format and never translated into English because translating big chunk of data in another language has risk of losing the real sense/meaning [64]. However, the themes or codes generated from the analysis were translated into English because those were shorter (in phrases or few words), easy to translate and less chance of losing the sense/meaning.

To facilitate the data analysis process, memo writing was done in three phases–after each interview, during transcription and during coding. They included reflections on environment, participants' way of expressing their feelings and responses, quality of interviewing, lessons learned and strategies for further interviews. Memos during transcribing focused on key words, patterns questions, and gaps to be filled. Memo writing during coding focused on the relationship between the key ideas and themes that emerged during the analysis.

Coding is the heart of analysis in grounded theory which is a process of sorting and synthesizing the large amounts of data into concise and meaningful terms by developing categories and sub-categories. It includes at least two main phases–initial coding and focused coding [61]. Initial coding (also called open coding) is the first phase of coding in which every line or words or segment of data is given a specific name. Focused coding (also called selective coding) is the second phase of coding in which the most significant or frequent initial codes are emphasized to sort, synthesize, integrate and organize large amounts of data [61].

We started initial coding by adopting the strategy of 'line-by-line' coding. Chunks of data were examined and granted a code for each line and/or segment from the transcript. Each code was assigned in a way that categorized, summarized and accounted for each piece of data. Gerunds (action verbs with -*ing* but used as a noun) and in-vivo codes (the same words that the participants expressed while interviewing) were used while generating codes. We aimed to avoid bringing our own views to coding which we tried to reflect data and the real experiences of participants. To ensure the codes reflect the meaning and experiences of the participants, we asked few participants who had prior experiences in research to review the initial codes. The codes were cross checked by the other authors to ensure that the pre-existing categories were not reflected during coding and development of tentative categories.

Focused coding started with the codes that were derived from initial coding. We chose those codes which appeared more frequently during initial coding or those which were more

significant in making analytic sense to categorize the data. We examined whether the selected codes matched with the whole data set and then compared between the codes. This comparison helped in identifying the codes having analytic power. Once the focused codes were developed, we compared them with the context, incident, and data to generate tentative categories. Tentative categories were developed from the codes that carried the most weight and reflected the ideas, events, or processes that the participants experienced. The categories were provisional in the beginning to remain open to further analysis.

Constant comparison was done within and between the data sets (interview statements and codes developed). First, we compared the interview statements for the same participant (since there were multiple interviews with each participant) and between different participants with similar subject matter and circumstances. The memos written during data analysis (analytical memo) were sorted, diagrammed and integrated to refine theoretical links [65]. The categories were compared, and a conceptual map was created to visualize the emerging theory. Subsequently, the analytical memos were integrated to contribute to the emerging theoretical framework.

## Ethical consideration

Ethical approval was received from Flinders University, South Australia on 3rd May 2018 (Ref. No. 7939), and from Pokhara University, Nepal on 7th June 2018 (Ref. No. 195.074/75). Permission was also received from the Ministry of Health, Nepal on 30th May 2018. Further modification on ethical approval was received from Flinders University on 4th December 2018 and from Pokhara University, on 16th December 2018. Participants were provided four sets of ethics documents prepared in local language–*Letter of Introduction*, *Information Sheet*, *Consent Form*, and *Approval Letter from the MoH*. Informed written consent was received from the participants prior to data collection and their anonymity and confidentiality was maintained during interviews. Participants were asked not to name anyone during the interviews. However, names were mentioned occasionally which were removed during transcribing. In those few interviews conducted at participant's workplace, some incidental people showed up which was carefully handled by pausing the interview and recording until there was complete privacy. Pseudonyms were used in the storage of data and in final reporting.

## Findings

This study involved 46 in-depth interviews with 22 public health officials. Participants were working in different institutions under the Ministry of Health with a range of work experience from local (district) level to federal (central) level. The key characteristics of the study participants are detailed in Table 1.

With reference to the socio-cultural context of Nepal, the grounded theory developed from this study describes the ways in which an ordinary individual becomes a leader in public health (Fig 1). Development of leadership within an individual is categorised and descried in four phases:

a. The initial phase in which an individual wishes to be or acts as a leader without a formal role or acknowledgement, where the family/social environment and individual characteristics play a role in influencing the actualisation of leadership styles and actions.

b. The identification phase in which an individual is formally/professionally identified and initiates their leadership journey in public health organisations.

c. The development phase in which an individual creates core leadership capabilities.

Table 1. Participants' profile.

| Characteristics | Classification | Size |
|---|---|---|
| Age | 30–40 years | 4 |
| | 40–50 years | 6 |
| | 50–60 years | 12 |
| Gender | Male | 16 |
| | Female | 6 |
| Academic qualification and background | Public health | 12 |
| | Medicine to public health | 6 |
| | Nursing/others to public health | 4 |
| Positions | Public Health Officer | 4 |
| | Public Health Administrator | 8 |
| | Regional/Central Directors | 10 |
| Total experience in health services | < 10 years | 6 |
| | 10–20 years | 3 |
| | 20–30 years | 7 |
| | > 30 years | 6 |
| Experience in leadership position | < 5 years | 12 |
| | 5–10 years | 3 |
| | > 10 years | 7 |

d. The expansion phase in which an individual expands their leadership capabilities and recognition.

## Phase 1 –initiating the foundations of leadership development

This is the earliest phase of leadership development process in which an individual configures, summons and enables the essence of leadership abilities because of their distinctive personal characteristics, and the family and social environment in which they grew up (Table 2). This phase mostly describes an individual's exposure to social context of leadership during childhood and adolescence. However, the components of this phase continually influence the process of leadership development throughout the life of an individual. During the time when family and social factors play a role in developing or emergent leadership, an individual knowingly and unknowingly observes the activities and characteristics of other people in society who have leadership roles. From that observation, an individual understands the nature and essence of leadership behaviour and leadership qualities to some extent. This understanding acts as a reference for leadership behaviours in the future.

**Individual characteristics.** Individuals who were successful in leadership naturally engaged with others (social mindset) and showed interests and involvement in social activities during childhood/adolescence. Because of their extrovert nature, they enjoyed connections with professionals and public. In addition, individuals who described themselves as stubborn were also successful in leadership because of their personal determination and an attitude to show others their capabilities.

*I was involved in many social activities during my adolescence. I enjoyed talking with people, helping them farm. . .. if I was in my village, people used to say that they don't need to be stressed. [P19]*

*. . .. I was a teen when I got married, I didn't know anything. . ..my husband's family decided not to educate me further. It was time to go to 'maiti' (mother's house) for cultural purposes*

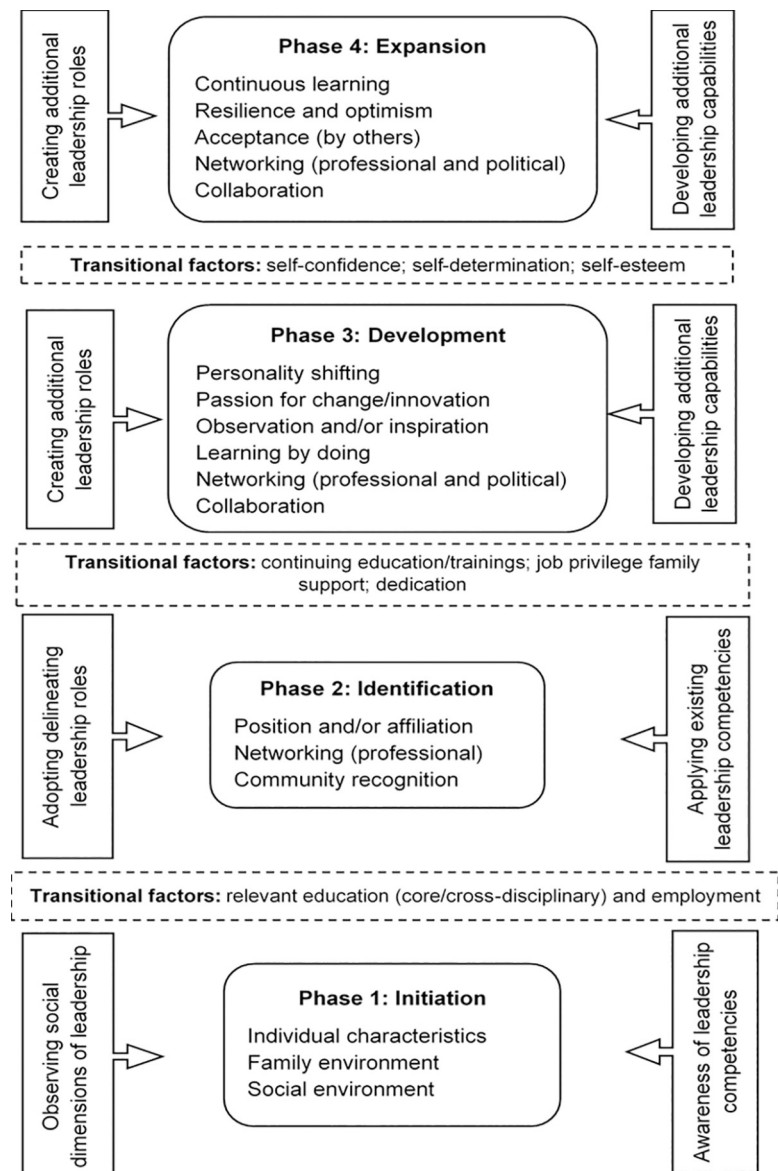

**Fig 1. Theory on public health leadership development in Nepalese context.**

**Table 2. Factors associated with the leadership development during childhood and adolescents.**

| Components | Sub-components |
|---|---|
| Individual characteristics | Personality<br>Social mindset |
| Family environment | Role of father and/or spouse<br>Gender behaviour<br>Socialization and social status<br>Economy |
| Social environment | Gender norms and roles<br>Cast culture Societal pressure |

*after marriage. . .. I told them that I will not return. . . .. . .I was not aware of all these things at that time . . .. that was disobedience. . .. But I did that for my study. [P12]*

**Family environment.** Family was found to be the first and most leading institution in developing leadership insights for the future, no matter whether the individual had this intention. The role of the father and/or spouse, gender behaviour inside family, socialization and the socio-economic status of the family were important to sensitize an individual towards leadership. The father acted as career anchor as well as the primary source of guidance for their children whereas the role of the mother was not obvious in terms of career progression. The father's support before marriage and the husband's support after marriage was a key to success among females. The role of the wives was important for their husband because they managed the household chores (one of the social norms in traditional Nepalese society) and accepted the external movement of their husbands for their academic and professional achievements. Gender equality inside the family provided a more equal opportunity for both males and females for their career development. Empowering girls from childhood helped them to develop dedication and commitment, by which they were able to tackle the unfavourable gender norms in society.

*To achieve this position, my wife played an important role because she provided me the freedom that I needed. I spent my family time with my clients. If she had have resisted this, it might not happen. [P15]*

*My parents never taught me that being a daughter I should not do this. . .. I didn't feel any difference since from my childhood. I had been treated equally like my brothers. I was motivated to sustain myself. . .. independently. . .. I think that encouragement had a positive count.* [P21]

Socialization during childhood determined the way an individual perceives people, resources, and services they lead. Individuals have life-long impression of their family's positive role and support to society. Individuals who grew up in this culture developed a 'culture of support' in their organisation and demonstrated the good value of resources.

*. . ..to be sensitive while providing services. . .I learnt this from my family. The nomads and poor people used to live in our spare house during their travel, and they got food from us. I learned to help people from my childhood. Even now, if I hear that some clients return without receiving services, I feel very bad. That service and this service (public health) is different, but the concern is same. We should provide whatever they (people) need.* [P3]

Economic resources were influential for education, which is a pathway for leadership. Families adopted need-based utilization of resources or even compromises with financial aspects to manage the expenses in education. Individuals who grew up in a culture that value for resources tried to manage the hardship by being self-reliant or by adopting alternative ways. Individuals who struggled with financial crisis/limitations, eventually developed financial consciousness.

*I usually didn't have lunch in the school's canteen because that money will be sufficient for me and my brother at home. Even if I thought to buy a book, the expenses for the whole month (for groceries) became inadequate. There were so many days that I walked without any money in my pocket. After my brother started teaching in school which made us a bit better.* [P3]

**Social environment.** The most influential social factors in developing individuals' career were gender norms and roles, caste/ethnicity culture and societal pressure. The traditional gender norms and roles in Nepalese society acted as an advantage for males but provided obstacles for females. Socially assigned additional responsibilities for women (such as care for children/family, household chores) and other cultural restrictions on them (such as staying at home and not being involved in social leadership) affected their access to education and/or jobs. This led to more critical self-perception and harmed leadership development. However, more gender equality in the family and self-dedication of an individual outweighed traditional gender norms.

*Obviously, it was a privilege as a male. I was free, I entertain the freedom as a male guy. I didn't face the barriers that a female had to face. That freedom, of course, helped me in my career development. [P7]*

*Our culture and norms benefited us (male), and. . .. yes. . .. it's difficult to act as a female. For example, I work and even my wife works but when we meet at the home in the evening, I expect a cup of tea from my wife. . .that means. . .our society still has this type of expectation. [P14]*

Socially assigned roles for certain caste/ethnic groups had established the values towards education, employment, and economy. Caste that valued education, emphasized achieving knowledge at any cost. Because of this education-preference culture, individuals from those caste were motivated to pursue education from childhood. This culture continued throughout their life and helped them to get better job opportunities. Eventually, it directed them into leadership positions.

*The characteristics inside a caste or the culture of caste. . ..is important. For example, our caste has a value to educate children. . .like as. . .. we have a culture of savings. We were educated from the ground reality that what to do, what not to do. So, it's not the caste but the culture of the caste that brought us in-front on job and education. [P15]*

Since Nepalese society judges the success and failure of individuals in terms of education and financial status, individuals constantly confronted moral pressure to show their achievements to others. This societal pressure acted positively to achieve better education/job and prepared them for future leadership.

*We . . . who are from a middle class or lower middle-class family. . .always have a pressure that 'you have to do something'. . .it is created by the society. I think, this social pressure worked for me. That pressure is painful sometimes but . . ..it is good for development.* [P15]

**Transitional factors between phase 1 and 2.** To proceed to the second phase of leadership development, an individual requires relevant education and a job. Formal courses in public health were the most common means of starting the early career in public health. The reasons behind enrolling in public health education and/or job were basically due to the following circumstances: alternative to a medical degree, interdisciplinary linkage between public health and other health professions, exposure to public health activities, and observing others.

**Phase 2 –getting identification and affiliation as a leader.** The second phase of leadership development starts when someone is identified as a public health official because of their position and affiliation with a health-related institution, networking with other professionals and recognition from the community. During this phase, individuals practise leadership based

on the roles and responsibilities assigned by the affiliated institution. For this, they utilize existing leadership competencies learned from educational courses as well as from the social experiences in the past.

**Position/Affiliation.**   The formal journey on public health leadership starts after obtaining an official position, such as public health officer or public health administrator. Being a chief executive of an institution, an individual commences leadership behaviour. Utilizing one's knowledge and expertise for better health outcomes by mobilizing staff and community people was considered as the core part and continuous job for public health officials.

*I was connected with community people and their health since I practiced as an Auxiliary Health Worker. After I became a public health officer, I started to think and act more from public health perspectives. [P1]*

**Community recognition.**   Through involvement in public health interventions and acting as a health educator in a region or locale, an individual gets recognition from the community. The level of recognition increases once the community feels there has been an improvement in services and indicators (visible outputs) such as availability of uninterrupted basic health services, and improvement in maternal and child health.

*Because of my public health activities, community people recognized me. . .it was such a backward community. . .. some changes provided me with satisfaction. Children started to get immunized; deliveries were conducted at health facilities. . .. though it was struggling, the results made me motivated. [P3]*

**Professional networking.**   Working as a public health official also increases connections with senior public health professionals and leaders because of which individuals develop professional networks. Continuous efforts to improve public health programs and improving health indicators results in increased status and recognition within the professional network and inside the government system.

*I developed more connection . . .. a type of relationship. Later, I received responsibility to handle the neonatal programs. . .. the mortality and morbidity decreased. . .. I became more recognized by my colleagues and seniors. [P9]*

**Transitional factors between phase 2 and 3.**   To proceed to phase 3, an individual needs additional education and/or relevant training to facilitate the leadership journey by updating/upgrading existing knowledge and skills. Continuing education is a means of career development and helps to enhance knowledge and expertise in leadership. Family support is also critical, as it needs to continue, to facilitate the longer leadership journey. Individuals who do not receive family support, need greater self-dedication and determination to proceed on their leadership journey.

## Phase 3 –developing core leadership capabilities

This is the most important phase of leadership development in which individuals practise more leadership than assigned (as in second phase) because of experience. While doing this, they realize their insufficiency of existing capabilities for leadership and attempts to develop additional skills and competencies. The factors contributing to leadership development in this phase are personality shifting, passion for change, observation/inspiration, learning by doing, and networking and collaboration.

**Personality shifting.** Public health officials, who defined themselves as quiet and shy, realized that public health needs extraversion characteristics. Exposure to public health activities and connection with other professionals helped them to develop extraversion characteristics that fit the nature of public health. This improved the way leaders speak, share, and communicate with others. Thus, an extrovert personality is one of the triggers for leadership development.

*I was shy at the beginning . . . I thought that I don't need to say, they (supervisors) will understand. But slowly, I changed myself. I started to speak. . .realized that I must talk. I felt that was a part of leadership development within me. Now, I know how to deal with seniors and how to make my view influential or which way to proceed. [P3]*

**Passion for change and innovation.** Individuals who wished to make an impact on policy and governance preferred to work with bureaucracy. Public health officials working in the non-government sector switched to work in the government sector after they realized that small things could bring a significant impact, if that could be done inside the structure of the health system. This passion for change and innovation encouraged them to challenge the improper functioning of the health system and discovered new ways/techniques to improve the health system and/or indicators. Experience acted as a backup for the execution of passion for change.

*I was working at the developmental sector (NGOs/INGOs) before coming here. Life there was so guided and structured. . .. . .couldn't implement changes or think differently. From there. . . I felt like . . . if I could be a part of government, then I can make an effort for changes. [P9]*

**Observation and inspiration.** Individuals observed the administrative and leadership performance of senior leaders and attempted to replicate them in their life. The things that the junior officials mimic from their seniors were their commanding capacity, decision making strategies, respect to work, relationship with staff, tactics for motivating staff, encouraging and convincing staff for additional responsibilities, and way of handling the diverse workforce and their expectations. However, a matching mindset between leader and follower was important for inspiration.

*. . .. . .his capacity for administration, capacity to feel public health, thoroughness, leadership capacity. . .. . particularly the multi-dimensional type of personality. . .I got inspiration from that. [P1]*

Apart from people, community acted as a subject of inspiration and learning for some officials. Observing community lifestyles, issues and problems, the way of managing resources, and their effort in struggling with scarcity and remoteness inspired them to serve their community much more. Providing health services was considered as a prestigious and religious job for some officials.

*When I feel alone or feel like I can't do anything, then I go to the field. . .mostly remote areas. I observe the community, their living status, their behaviours. After I see people struggling with hardship, I realize how lucky I am, and still, I have so many things to do. This thing charges me, I get energy . . . for a whole month from a single visit to community. [P3]*

**Exposure and experiences.** Public health officials learnt to lead from their experiences even from the time when they worked as a basic health worker. Despite of minimal trainings/courses related to leadership, prolonged engagement with community and continuous experiences in public health activities provided them an opportunity to understand leadership application. Experiences resulted in the better understanding of needs and issues of institutional staff and community people, thus public health officials were able to handle a diverse range of workforce demographics and rising public expectations. Experience was also a factor for leadership maturity.

*While working as institution chief. . .. most of the time was spent on leadership and decision making. In the beginning, I made immature decisions and inappropriate leadership. But as I worked more, I learned many things. . .my learning continued. . .. I applied practical things from my study. . .. I became mature as the time passed. [P20]*

Exposure in informal gatherings was an additional advantage for public health officials as the discussions in those situations meant individuals have a sequence for official decision making in the future. However, this was a barrier for female officials who were constrained by traditional gender norms and roles. Because of this, female officials were deprived of social learning (via informal gatherings) and their chances to get additional responsibilities. However, females who were conscious of their leadership development and who challenged the traditional gender norms, adopted a strategy to balance their work behaviour.

*As a female, I can't stay till midnight. . .. . .and can't drink (liquor) as well. There are various reasons that I must be home on time. [P10]*

*After coming here (Ministry of Health), I changed my strategy. I go wherever they invite me. I know my limitations, it's not necessary to get drinks (liquor). I participate as far as practicable by arranging the time and convincing my family. Now, they call me in to policy level discussion. I know. . . it's not easy for all ladies. . .. they are bounded. [P21]*

**Networking and collaboration.** Being a public health official, individual works collaboratively with both health and non-health sectors. This sort of networking and collaboration demands advocacy and communication, the practice of which helps in developing leadership skills. Apart from the second phase (initiation) in which a public health official mostly engages in professional networking, the third phase involves networking and collaboration with political leaders and non-health ministries. With this, public health official develops additional leadership skills such as negotiation and lobbying.

*For that (policy), my and the minister's wavelength matched. I was planning and designing the program, and he had the intention to do that. I was advocating for him, and he asked me to generate evidence. [P9]*

**Transitional factors between phase 3 and 4.** During the third phase, individual obtain an acceptable level of leadership recognition based on their position as well as perceived by their staff. They continually strive to develop their leadership capabilities and wishes to expand their leadership status beyond the public health sector. Individuals require self-confidence, self-esteem, and self-determination to accept the leadership challenges and to motivate self to go above and beyond.

### Phase 4 –expanding leadership capabilities

This is the final phase of leadership development process in which individuals expand and increase their leadership abilities by continuous self-directed learning, resilience, and optimism, and by broadening professional and political networking. In this phase, a public health official reaches the highest level of leadership by position, and most importantly by skills and experiences. This makes an individual able to modify (or even manipulate) leadership roles and strategies to bring in desirable changes to the health system. This phase is critical for every public health official as they struggle to retain their status of acceptance (by others) and social prestige received as a part of their leadership journey.

**Continuous self-directed learning.** Being studious and updated with contemporary social, political and public health issues, public health officials maintained the superiority of their leadership and also led the newcomers with their revitalized knowledge. Public health officials who were eager to learn, created an enabling environment for themselves.

*Learning is like. . .. I can learn from supervisors, even from an office assistant. . . if you have the intention to learn or if you are positive to adopt learning. . .. . .there is environment to learn. . ..in fact, you create the environment by yourself. [P14]*

**Resilience and optimism.** After having intense experiences of leadership, public health officials understood the value of human resources and the reciprocal relationship with them. Being resilient and optimistic was one of the accomplishments of their leadership, which they used to mobilize their team the way they desire.

*Now I feel that I had walked in the right path. My past works are motivating me. . .. whatever I initiated in the past, that is in practice nowadays. . .. I am satisfied. [P19]*

**Acceptance (by others).** Public health officials maintained excellent relationships with others because of their loyalty in providing services, dedication to enhance health system and practising justice and equity in the workplace. This led them to be accepted by their staff as well as by the professional and political alliances. Advocacy, lobbying and negotiation with political stakeholders and ministerial were considered as a must to prevent and control undue influence in leadership practices.

*All Directors are assumed to have political affiliation. Although, people believe that I am also affiliated to one of the political parties, I am not experiencing difficulties from that. Still I believe, I am well accepted by all political parties. [P18]*

## Discussion

Leadership development theories explore and describe how leadership emerges throughout an individual's lifetime. Leadership development has been defined as "almost every form of growth or stage of development in the life cycle that promotes, encourages, and assists in one's leadership potential which includes all types of formal and informal learning activities from childhood development, education and adult life experiences" [66]. This definition is confined to the individual perspectives of developing leadership capabilities. However, there are other literatures that describes the same concern using the term 'leader development' [44,50,56,67–

70]. Leadership development in this study is about exploring the factors that individuals exposed to during their lifetime and because of which they were positioned and identified as a leader in public health. Since it was not possible to understand leadership development without exploring the life histories of leaders, we concerned to fully contextualise leadership development in the life cycle.

The grounded theory developed from this study shapes leadership in public health as a journey and the result of multiple factors initiated from childhood/adolescence, which developed gradually as the individual exposed to public health activities. The foundation of leadership starts in the early stages of life when the individual characteristics, family and social factors influence the way individuals observe, understand, and make sense of the prerequisites for leadership. Family factors such as role of the father, gender equality, socialization, family status and economy were found to be important among the participants in this study that supported them in accessing and completing formal education as well as in motivating and facilitating them continuously for personal development. This study also explored that individual with stubborn and assertive personalities were able to create their career path in public health because of their 'ego' to show others their capabilities. Studies revealed the experiences during childhood and adolescence influenced the personal characteristics such as assertiveness, confidence and need for achievement [71] as well as attitude and behaviour towards leadership [44]. Among the four agencies of socialization (family, schools, peer groups and mass media) [72], participants in this study were mostly influenced by family members, particularly their fathers. Studies have shown the importance of family members including the exceptional role of the father in building confidence, positive parenting, and family life with strong ethics in successful leadership [42,43,50,73,74]. It is also believed that socialization during childhood lasts longer [75] and influences the way an individual makes sense of their organisational environment after they are employed [76]. As an example, participants from this research who suffered financial hardship during their childhood/adolescence were found to be more sensitive towards management of organisational resources and the services provided to the public.

Social and cultural contexts are considered important in leadership development among individuals [50]. During the time of this study, more than two-thirds of public health officials in Nepal were from the so-called upper castes/ethnicities such as Brahmin, Chhetri and Newar [77]. Ethnicity in Nepal is not just a person's race; it is about institutions, learned behaviour and customs [78]. Caste position was a critical issue in the past [79] and caste relations remain a predominant system of social stratification and inequality [80]. Since Nepal has a long history of marginalization and exclusion from access to resources, services and opportunities, discrimination and marginalization based on caste/ethnicity, gender, and geographical remoteness, most achievements in the past have gone to the governing caste and/or ethnic groups [78]. Research participants from those ethnic groups accepted the positive role of caste culture in such a way that the culture inside the caste increased their access to education and encouraged them to obtain employment in the government sector, thus positively favouring their leadership opportunities.

Gender was another prominent social variable explored in this study that influenced the process of leadership development. The secondary data collected for the purpose of this study showed the number of female public health officials being extremely small (10.6%) compared to the number of men (89.4%) [77]. This evidence suggested that opportunities for leadership and career advancement are very different for women than for men. Male participants in this study were privileged by their family and society to stay outside home for education and/or employment opportunities. Despite socio-cultural barriers and gender inequality, female officials had unconditional support from their family (especially their father and their husband) and dared to accept the challenges. As a result, they were able to represent their gender in

leadership positions. The reason behind poor representation of women was mostly due to the wider, patriarchal nature of Nepalese society which is well documented as one of the systematic barriers to gender equality in the country [81]. Gender equity, personal values, persistence, and self-awareness were some of the important aspects of leadership development in females [82] for whom gender stereotypes influence their underrepresentation in leadership positions [83].

Being positioned as public health official in formal organisations as well as being involved in community activities makes an individual more likely to be identified as a formal leader in public health. Since leadership is difficult to enact solely, there is need of affiliation with organisations to develop capabilities, connections, system and culture [67]. Public health professionals usually begin their leadership journey as a team member, which gradually progresses to leadership positions to influence and motivate others [39]. This study explored that leadership development is more than being positioned. Multiple factors played a role in developing leadership capacity such as shifting of personality, passion for change, observation and inspiration, exposure and experiences, and networking and collaboration. A study in India and China (neighbouring countries of Nepal) showed that factors responsible to improve leadership skills among managers were–supervisors who were role models, coursework and training, working with difficult people, feedback and coaching, employing new initiatives, personal experiences and stakeholder engagement [44]. Observation/interaction with others [73], support from seniors [84] from the same field as well as regular contacts, connections with high-level interactions with people from outside helps in preparedness for and access to leadership opportunities [85] and developing leadership skills [44]. Engagement in informal gatherings was challenging for female leaders from this study because of the socio-cultural restrictions and traditional gender norms. This results lower social capital among them and further limits their prospects for interconnections [86], which in turns, affects their access to leadership opportunities. Participants who were passionate for change and innovation believed in increasing of their leadership status inside the bureaucracy as well as within their professional circle. Doing innovative work also helped the Indian and Chinese managers to foster their leadership [44].

The role of training, workshops and other formal/informal courses in leadership were highly emphasized as being effective in public health leadership [34,51,87–89]. In contrast to this, participants from this study valued 'learning by doing' in developing leadership capabilities. All participants mentioned that they learned most of their leadership skills via exposure/ experiences to public health activities and additional responsibilities. Experiences was also found to be an important aspect of leadership development in various studies from non-health sectors [42,44,57,66,73,74,84,90,91]. Since leadership is developed through the enactment of leadership, the role of experiences in leadership could not be ignored [92]. It is also argued in the literature that traditional schools and programs in public health are not enough for public health professionals to develop practical leadership skills [93,94], hence experiences matter more among public health officials in Nepal. Developing leadership is a lifelong journey in which an individual learns lessons from experiences [42,57] and constructs meaning from those experiences [44,91]. Engagement in more leadership positions develops greater leadership efficacy and confidence to lead [73] and exposure to difficult and challenging tasks is the best opportunity for professional growth in leadership [16]. Experiences and career advancement help leaders to tackle emerging challenges [56,95], which is more meaningful in the public health context where leaders need to lead continuously despite the uncertainty and complexity of public health threats.

The theory indicates that developing leadership capabilities and being recognized by bureaucracy and/or the health system does not complete the journey of leadership development. Public health officials with their self-confidence, self-determination, and self-esteem,

continually strive to increase their leadership capabilities and believe they can move ahead in formal leadership positions/roles. A grounded theory on leader identification development among college students also explored that as the time passes, individuals develop self-awareness, self-confidence, and interpersonal efficacy and became self-directed without others' support [42]. Similarly, more effective leadership occurs though continuous learning [96] and self-development of individual leader [57]. Emphasizing self-development, public health officials in this study practised resilience and optimism and were accepted by others as public health leaders.

The components of grounded theory from this research resonate with the work on leader development presented by some scholars [42,43,56,66,97]. The theory developed from this study is different from the existing theories because it is grounded in primary data and describes leadership development through the life-history perspectives of individuals. Apart from other models/frameworks in leader or leadership development which focused on the role of either child/adolescent development [43] or adult development [56,57], this study explored the role of both childhood and adult development in leadership. This study adds a new theory in the field of leader/leadership development where further research were recommended [50,57,66,68]. Most importantly, this study laid a foundation to understand the public health leadership in the context of developing countries.

## Strengths and limitations

A preliminary literature review was done prior to the study and the existing knowledge gap in public health leadership was identified. Because of this, grounded theory methodology (GTM) was adopted to construct a leadership development theory in public health. Careful application of processes and methods of GTM ensured the originality, credibility, resonance, and usefulness of the developed theory. This study included the participants from each level and position, working from district to central level, as well as from different localities, castes/ethnic groups, gender, and educational background. This has ensured diversity in the sample. The theory is applicable in the Nepalese context and could also be generalized in other countries having similar socio-cultural characteristics, especially the countries in South-East Asia. The major limitation of this study is the selection of participants from government institutions based on their existing leadership positions who were preidentified leaders in public health. This might have missed potential participants who had leadership capabilities but were not engaged in leadership positions or who were working in non-governmental sectors. To overcome this limitation, we selected some participants who already had experience in non-governmental sector and explored the ways of leadership development over there. The theory was validated by interviewing key participants (former and current public health leaders and pioneers) in the government, non-government, and academic sector. This confirms the findings reflect the overall scenario of public health leadership in Nepal.

## Conclusions and recommendations

The grounded theory developed from this study suggests that leadership in public health is a journey and the result of multiple factors initiated from childhood/adolescence that grows, changes, and increases as an individual experiences public health activities. It also captured insights into the meaning and actions of participants' professional experiences and highlighted the role of individual characteristics, family and socio-cultural environment, and workplace settings in the development of leadership capabilities. It offers new concepts on leadership development (such as how someone become a leader in public health) and contributes to fulfill the existing knowledge gap in literature. The theory can be used as a conceptual framework for

other qualitative and quantitative studies in public health and other close disciplines like medicine and nursing. This study compiled individual factors to develop a grounded theory, however, all the individual factors may not have intense exploration. Therefore, further studies are recommended to explore the leadership dimensions by using individual factors such as gender, caste culture, education, experiences, and self-learning. This will broaden the theoretical base of public health leadership.

## Supporting information

**S1 Text. Interview guidelines.**
(PDF)

**S2 Text. Example of a memo writing.**
(DOCX)

**S3 Text. Examples of coding.**
(PDF)

## Acknowledgments

We express our sincere thanks to the Ministry of Health (MoH)–Nepal, Nepal Health Research Council, Pokhara University–Nepal, and Flinders University–Australia for their administrative support. We are grateful to the public health academics and officials in Nepal including Prof. Madhusudan Subedi (Patan Academy of Health Sciences), Dr. Binjwala Shrestha (Institute of Medicine), Dr. Mahendra Sapkota (Kathmandu University), Dr. Baburam Marasini (Ex-Director, Department of Health Services) and Mr. Shree Krishna Bhatta (Ex-Director, Regional Health Directorate) for their technical support during this study. We are also thankful to all the research participants for their contributions in this study.

## Author Contributions

**Conceptualization:** Sudarshan Subedi, Colin MacDougall, Darlene McNaughton.

**Formal analysis:** Sudarshan Subedi, Colin MacDougall, Darlene McNaughton, Udoy Saikia, Tara Brabazon.

**Investigation:** Sudarshan Subedi.

**Methodology:** Sudarshan Subedi, Colin MacDougall, Darlene McNaughton.

**Writing – original draft:** Sudarshan Subedi.

**Writing – review & editing:** Colin MacDougall, Darlene McNaughton, Udoy Saikia, Tara Brabazon.

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
