## [Decision Letter · Decision Letter 0]

18 Aug 2021

PONE-D-21-08622

Leadership development among public health officials in Nepal: A grounded theory

PLOS ONE

Dear Dr. Sudarshan Subedi,

Thank you for submitting your manuscript to PLOS ONE. After careful consideration, we feel that it has merit but does not fully meet PLOS ONE’s publication criteria as it currently stands. Therefore, we invite you to submit a revised version of the manuscript that addresses the points raised during the review process.

We look forward to receiving your revised manuscript.

Kind regards,

Sharon Mary Brownie

Academic Editor

PLOS ONE

Editor Comments

The review process has some areas where your work can be improved. Please pay careful attention to each revision recommended. Please also seek professional assistance in respect to English gramma and spelling/ You may find it useful to seek the help of a professional editing service.

Journal Requirements:

2. In the Methods section, please consider including more information on the number of interviewers, their training and characteristics; and specify whether an interview guide was used to interview the participants in your study. Furthermore, please provide additional information regarding the interview guide development process, including the theories or frameworks which were employed. 

3. Please provide additional details regarding participant consent. In the ethics statement in the Methods and online submission information, please ensure that you have specified (1) whether consent was suitably informed and (2) what type you obtained (for instance, written or verbal). If your study included minors under age 18, state whether you obtained consent from parents or guardians. If the need for consent was waived by the ethics committee, please include this information.

6. We note you have included a table to which you do not refer in the text of your manuscript. Please ensure that you refer to Table 1 in your text; if accepted, production will need this reference to link the reader to the Table.

Reviewers' comments:

Reviewer's Responses to Questions

**Comments to the Author**

1. Is the manuscript technically sound, and do the data support the conclusions?

Reviewer #1: Yes

Reviewer #2: Yes

Reviewer #3: Partly

2. Has the statistical analysis been performed appropriately and rigorously? 

Reviewer #1: N/A

Reviewer #2: I Don't Know

Reviewer #3: N/A

3. Have the authors made all data underlying the findings in their manuscript fully available?

Reviewer #1: Yes

Reviewer #2: Yes

Reviewer #3: No

4. Is the manuscript presented in an intelligible fashion and written in standard English?

Reviewer #1: Yes

Reviewer #2: Yes

Reviewer #3: Yes

5. Review Comments to the Author

Reviewer #1: Provide a definition of leadership upfront in the background

You mention that 'These officials are not formally employed as leaders but regularly exercise or enact leadership in their work. Stronger leadership in public health is needed to accelerate improvement in public health and this will happen when the public health practitioners have commitment and competencies to perform leadership...'

What are the actual gaps in their leadership? can you state these in that paragraph.

The rationale for leadership development is good

Were there any potential ethical issues with the focal person contacting the participants on behalf of the study team. Is it possible that some of the participants felt obliged to participate?

Please clarify what you mean by 'Exposure to incidental people during interviews was carefully handled.'

Results are well-presented

Were there any differences by gender or years of experience in the perspectives?

Good paper overall

Check for grammatical and typographical errors. For example, 'Literatures recommended' in abstract

Reviewer #2: The manuscript technically sound also data support the finding.

The analysis of the interview data were done in Nibali language which is good to reflect the real views of the participant but the authors must explain how do they translate those data or the codes to English? if they do so. I am satisfied with the writing fashion and English standard.

Reviewer #3: This is a qualitative study of leadership identity development in the Nepalese healthcare system. It brings interesting and well-coded information about how a representative and diverse group of managers have experienced coming into these roles, spanning castes, gender and professions. I am mmuch in favor of these kinds of studies and think this is very promising. However, I still do not think this study is finished in its present form. It is under-developed conceptually and the text suffers from it.

First of all, the focus and contribution is insufficiently conceptualized and not sufficiently grounded in literature. The study claims in the beginning and in the discussion section that a "gap" in the literature is identified, but I find this hard to follow in the present form. For example, a search in Web of Science using the two key words "leadership" and "healthcare" renders 8,800 publications. Adding "developing countries" as a key word, there are still 350 publications to review. The study needs a clearer focus and aim, relating the questions asked and the data collected to a more narrow set of research questions.

This unfortunate situation creates two quite strong flaws in the text. The first is that I, as a reader, am uncertain what the data actually represent. For example, are the stages of development that are presented something that were theoretically assumed a priori - a template to explore the subjects and their stories? Or were they derived from the interviews? I like to read the excerpts from the interviews but they do not inform a coherent set of research questions in my mind, and this is tricky for a scientific article. Another example is on page 28, which reads: "However, training and coaching were found to be negligible in developing leadership among public health officials in Nepal. Participants in this research had no or very minimal exposure to coursework and training related to leadership, consequently they emphasized more on 'learning by doing' or experience." I did not have the feeling that this was properly prepared and conceptualized prior to or during the interviews, and neither do I as a reader know what the subjects really said here. A stricter relationship between the questions and the results would be good.

The second becomes apparent in the discussion, where the text wanders a bit aimlessly about. For example, the role of genetic components in leadership talent is discussed but the study bears absolutely no data with relevance to this question. Such passages contribute to a feeling of lacking rigor in the conceptual foundation of the whole study.

In my opinion, the study would improve greatly by making more out of Nepal as a case in point for development of healthcare professionals as an important tool in the development of the services for the population. The lack of resources, the caste and gender issues, the groups of professionals and their relationships to their communities are important contexts to master for the system. I think that the authors do possess the necessary data to write an interesting study about nepalese healthcare officials but the present form is too narrow. I suggest that the authors reduce their ambitions to write something with general implications for leadership theories and maximize their focus on the local and developmental challenges, along with the practical consequences that follow for the professionals themselves and the authorities that employ them.

Finally, the text suffers from typos, incomplete sentences and ambiguous language. I think the authors should hve the manuscript proofread, possibly by a professional agency.

6. PLOS authors have the option to publish the peer review history of their article (what does this mean?). If published, this will include your full peer review and any attached files.

Reviewer #1: No

Reviewer #2: No

Reviewer #3: **Yes: **Jan Ketil Arnulf

---

## [Author Response · Author response to Decision Letter 0]

4 Oct 2021

Dear Editor,

On the behalf of our team, I would like to thank you and the reviewers for their valuable time in providing constructive feedback to make our manuscript suitable for publication. With careful attention to each comment raised by the editor and reviewer, we have corrected and uploaded a revised version of manuscript with additional supporting files. 

Comments from Editor and response to them

Comment: The review process has some areas where your work can be improved. Please pay careful attention to each revision recommended. Please also seek professional assistance in respect to English grammar and spelling/ You may find it useful to seek the help of a professional editing service. 

Response: Thank you for your feedback and we have paid close attention to grammar.

Comment: Please ensure that your manuscript meets PLOS ONE's style requirements, including those for file naming.

Response: Headings, font type, font size and file name has been corrected as per the journal requirements. One of the authors' affiliations has also been corrected

Comment: In the Methods section, please consider including more information on the number of interviewers, their training and characteristics; and specify whether an interview guide was used to interview the participants in your study. Furthermore, please provide additional information regarding the interview guide development process, including the theories or frameworks which were employed. 

Response: More information has been added under 'data collection' section.

Comment: Please provide additional details regarding participant consent. In the ethics statement in the Methods and online submission information, please ensure that you have specified (1) whether consent was suitably informed and (2) what type you obtained (for instance, written or verbal). If your study included minors under age 18, state whether you obtained consent from parents or guardians. If the need for consent was waived by the ethics committee, please include this information.

Response: More information has been added under 'ethical consideration' section.

Comment: In your Data Availability statement, you have not specified where the minimal data set underlying the results described in your manuscript can be found. PLOS defines a study's minimal data set as the underlying data used to reach the conclusions drawn in the manuscript and any additional data required to replicate the reported study findings in their entirety. All PLOS journals require that the minimal data set be made fully available. For more information about our data policy, please see http://journals.plos.org/plosone/s/data-availability. Upon re-submitting your revised manuscript, please upload your study’s minimal underlying data set as either Supporting Information files or to a stable, public repository and include the relevant URLs, DOIs, or accession numbers within your revised cover letter. For a list of acceptable repositories, please see http://journals.plos.org/plosone/s/data-availability#loc-recommended-repositories. Any potentially identifying patient information must be fully anonymized. We note that you have indicated that data from this study are available upon request. PLOS only allows data to be available upon request if there are legal or ethical restrictions on sharing data publicly. For more information on unacceptable data access restrictions, please see http://journals.plos.org/plosone/s/data-availability#loc-unacceptable-data-access-restrictions. 

Response: There are not legal or ethical restriction on sharing data publicly. We have uploaded the interview guide, example of memo writing and example of coding as a part of manuscript. We haven't uploaded any data to public repository. Regarding the data set underlying the results (interview transcripts), we have stated that the data will be available upon request. This is because all the interview transcripts are in local (Nepali) language, not in English. Even if we translate the data, we will lose the meaning, and this inhibits the ability of participants to be heard. However, if the editorial team expects the interview transcript to be uploaded, we are happy to share some of the interview transcript in its original format by removing the private/critical information.

Comment: In your revised cover letter, please address the following prompts: a) If there are ethical or legal restrictions on sharing a de-identified data set, please explain them in detail (e.g., data contain potentially sensitive information, data are owned by a third-party organization, etc.) and who has imposed them (e.g., an ethics committee). Please also provide contact information for a data access committee, ethics committee, or other institutional body to which data requests may be sent. b) If there are no restrictions, please upload the minimal anonymized data set necessary to replicate your study findings as either Supporting Information files or to a stable, public repository and provide us with the relevant URLs, DOIs, or accession numbers. For a list of acceptable repositories, please see http://journals.plos.org/plosone/s/data-availability#loc-recommended-repositories.

Response: Please see the cover letter. We have clarified the issue of data availability in that letter.

Comment: We note you have included a table to which you do not refer in the text of your manuscript. Please ensure that you refer to Table 1 in your text; if accepted, production will need this reference to link the reader to the Table.

Response: Corrected for table 1. Please see under 'Findings' section

Comment: Please include captions for your Supporting Information files at the end of your manuscript, and update any in-text citations to match accordingly. Please see our Supporting Information guidelines for more information: http://journals.plos.org/plosone/s/supporting-information. 

Response: Done. Please see at the very end of the manuscript file.

Comments from Reviewer 1 and response to them

Comment: Provide a definition of leadership upfront in the background

Response: Done. Please see the first paragraph in 'background' section

Comment: You mention that 'These officials are not formally employed as leaders but regularly exercise or enact leadership in their work. Stronger leadership in public health is needed to accelerate improvement in public health and this will happen when the public health practitioners have commitment and competencies to perform leadership...'

What are the actual gaps in their leadership? can you state these in that paragraph.

Response: Done. Please see towards the end of second paragraph in 'background section'.

Comment: The rationale for leadership development is good.

Response: Thank you.

Comment: Were there any potential ethical issues with the focal person contacting the participants on behalf of the study team. Is it possible that some of the participants felt obliged to participate? 

Response: A sentence clarifying this issue has been added under the section 'recruitment of participants'.

Comment: Please clarify what you mean by 'Exposure to incidental people during interviews was carefully handled.'

Response: Done. Please see the section 'ethical consideration'

Comment: Results are well-presented.

Response: Thank you.

Comment: Were there any differences by gender or years of experience in the perspectives?

Response: In this study, we explored the factors responsible for leadership development, and 'gender' and 'years of experiences' were few of them. We did not explore the differences of leadership development by 'years of experiences' and by 'gender'. However, this paper has some useful information regarding the role of gender (see table 2 and its description in phase 1) and experiences (see the subsection 'exposure and experiences' under phase 3 findings) in developing leadership insights.

Comment: Good paper overall.

Response: Thank you.

Comment: Check for grammatical and typographical errors. For example, 'Literatures recommended' in abstract

Response: Proof reading has been done with an experienced editor.

Comments from Reviewer 2 and response to them

Comment: The manuscript technically sound also data support the finding. 

Response: Thank you. 

Comment: The analysis of the interview data was done in Nepali language which is good to reflect the real views of the participant, but the authors must explain how they translate those data or the codes to English? if they do so. I am satisfied with the writing fashion and English standard.

Response: Done. Please see the first paragraph under 'data analysis' section.

Comments from Reviewer 3 and response to them

Comment: This is a qualitative study of leadership identity development in the Nepalese healthcare system. It brings interesting and well-coded information about how a representative and diverse group of managers have experienced coming into these roles, spanning castes, gender and professions. I am much in favor of these kinds of studies and think this is very promising. However, I still do not think this study is finished in its present form. It is under- developed conceptually and the text suffers from it. 

Response: Thank you for the feedback and we are happy to know that you prefer this type of study. By addressing the reviewers' comments, we have further developed the paper conceptually.

Comment: First of all, the focus and contribution is insufficiently conceptualized and not sufficiently grounded in literature. The study claims in the beginning and in the discussion section that a "gap" in the literature is identified, but I find this hard to follow in the present form. For example, a search in Web of Science using the two key words "leadership" and "healthcare" renders 8,800 publications. Adding "developing countries" as a key word, there are still 350 publications to review. The study needs a clearer focus and aim, relating the questions asked and the data collected to a more narrow set of research questions.

Response: In this study, we examined 'public health leadership', and distinguished it from 'health care leadership'. Theoretically, we draw differences between public health leadership and health care leadership. The healthcare sector in Nepal usually refers to 'hospital and/or clinical setting' and consists of different health care professionals such as doctors, nurses, pharmacists, radiologists. Practically, the health system of Nepal has distinguished 'health care' and 'public health' sectors legislatively which we can see under the 'health services rules and regulations' (please see the supporting file attached as annex). For example, there are 'district hospitals' and 'district public health offices' in Nepal. The chief of district hospital is a clinical doctor, and the focus is on curative and rehabilitative services whereas the chief of a district public health office is a public health officer, and the focus of this institution is 'disease prevention and health promotion'. Similarly, there are regional hospitals and regional health directorate with different health professional and areas of work. We have clarified the distinction between public health leadership and healthcare leadership and described the background of participants and their areas of work. Now it has been corrected. Please see 'study settings' under methodology. We included the research question that guided the study. Thank you for pointing out this important thing. Now, it has been added. Please see the last paragraph of 'background' section. Regarding the literature, we did a narrative review by adapting the systematic procedures (Jan 2000 to Dec 2019) on 'public health leadership' prior to this study. Initially, we searched for 'public health' and 'leadership' in full text which was completely unmanageable (e.g. 446,071 articles in ProQuest and 21,048 in PubMed) which was searched again in another field except full text which was also unmanageable (47,354 in ProQuest). Containing these two terms in full text doesn't necessarily mean that the work is on public health leadership. Most of the papers include these terms as a part of providing introduction or discussion for other non-leadership research. So, we decided to search with the phrases 'public health' and 'leader*' in the document title. The rationale for searching the title was that if the study has complete focus on public health leadership, then the term 'public heath' and 'leader/leaders/leadership', should have been used in the title of articles despite their position. The initial searches on databases resulted in 767 articles. This number was then reduced by removal of extraneous articles through duplication (402), title relevancy (162), abstract availability and relevancy (116), and full text relevancy (40). After removing the extraneous articles, 47 remained for review. Bibliography tracing was then done from these articles and relevant references were searched in google and/or website of stated journal. This resulted in additional nine papers that were relevant to include. In this way, 56 articles were used for review. The third paragraph of the Introduction section has been revised to reflect this research gap.

Comment: This unfortunate situation creates two quite strong flaws in the text. The first is that I, as a reader, am uncertain what the data actually represent. For example, are the stages of development that are presented something that were theoretically assumed a priori - a template to explore the subjects and their stories? Or were they derived from the interviews? I like to read the excerpts from the interviews but they do not inform a coherent set of research questions in my mind, and this is tricky for a scientific article. 

Response: We did not used any existing framework to create the theory. This theory is completely grounded in data and was progressively developed from the data collected from the interviews. This has been made clear at the last paragraph of 'Background' section. 

Comment: Another example is on page 28, which reads: "However, training and coaching were found to be negligible in developing leadership among public health officials in Nepal. Participants in this research had no or very minimal exposure to coursework and training related to leadership, consequently they emphasized more on 'learning by doing' or experience." I did not have the feeling that this was properly prepared and conceptualized prior to or during the interviews, and neither do I as a reader know what the subjects really said here. A stricter relationship between the questions and the results would be good.

Response: Thank you for pointing this out. We have found that participants had exposure to general managerial/technical trainings as a part of job but did not get specific trainings/courses related to leadership development. This information was not included in the findings section because while describing our findings in terms of grounded theory, we preferred to mention what worked (e.g., learning by doing/experiences) rather than what didn't worked. To make it clear, we have discussed this issue in 'discussion' section by comparing our results with other studies. To address your concern, we have revised the specific section under findings (please see subsection 'exposure and experience') as well as in discussion section (paragraph 6).

Comment: The second becomes apparent in the discussion, where the text wanders a bit aimlessly about. For example, the role of genetic components in leadership talent is discussed but the study bears absolutely no data with relevance to this question. Such passages contribute to a feeling of lacking rigor in the conceptual foundation of the whole study.

Response: Thank you for pointing this out. We have revised the discussion section to address this. Please see the second paragraph of discussion section. 

Comment: In my opinion, the study would improve greatly by making more out of Nepal as a case in point for development of healthcare professionals as an important tool in the development of the services for the population. The lack of resources, the caste and gender issues, the groups of professionals and their relationships to their communities are important contexts to master for the system. I think that the authors do possess the necessary data to write an interesting study about Nepalese healthcare officials, but the present form is too narrow. I suggest that the authors reduce their ambitions to write something with general implications for leadership theories and maximize their focus on the local and developmental challenges, along with the practical consequences that follow for the professionals themselves and the authorities that employ them.

Response: We highly appreciate your feedback on this. As we aimed to develop a grounded theory on individual leadership development at local level, we have discussed the aspects that we got from the interviews. So, the practical application of leadership or leadership enactment is out of the scope of this paper. As you said, the socio-cultural issues and resource inadequacy are the important aspects of Nepalese health system, we have discussed those aspects that we explored from the interviews. For example, you can see the findings about the role of 'gender' and 'cast/ethnicity' in leadership development in this paper. You can see this in the description of "phase 1"under the subsection 'family environment' and 'social environment' with appropriate quotations. The summary of the components is also mentioned in "table 2". We have also discussed (and revised) the issues of 'gender' and 'cast/ethnicity' adequately. Please see 3rd and 4th paragraphs of discussion section. We hope you will understand the background and scope of this study and help us in fulfilling the literature gap in South-East Asia Region. We are hopeful that his study will act as a useful reference for future research at local level.

Comment: Finally, the text suffers from typos, incomplete sentences, and ambiguous language. I think the authors should have the manuscript proofread, possibly by a professional agency.

Response: Proof reading and editing has been done by experienced editor.

---

## [Decision Letter · Decision Letter 1]

18 Oct 2021

Leadership development among public health officials in Nepal: A grounded theory

PONE-D-21-08622R1

Dear Dr. Sudarshan Subedi,

We’re pleased to inform you that your manuscript has been judged scientifically suitable for publication and will be formally accepted for publication once it meets all outstanding technical requirements.

Kind regards,

Sharon Mary Brownie

Academic Editor

PLOS ONE

Reviewers' comments:

Reviewer's Responses to Questions

**Comments to the Author**

Reviewer #1: All comments have been addressed

Reviewer #3: All comments have been addressed

2. Is the manuscript technically sound, and do the data support the conclusions?

Reviewer #1: Yes

Reviewer #3: Yes

3. Has the statistical analysis been performed appropriately and rigorously? 

Reviewer #1: N/A

Reviewer #3: N/A

4. Have the authors made all data underlying the findings in their manuscript fully available?

Reviewer #1: Yes

Reviewer #3: No

5. Is the manuscript presented in an intelligible fashion and written in standard English?

Reviewer #1: Yes

Reviewer #3: Yes

6. Review Comments to the Author

Reviewer #1: All my previous comments have been addressed satisfactorily. The authors could however, do a language sweep to address any minor grammatical and typographical errors.

Reviewer #3: I find the revised version of this manuscript much improved. It as addressed all the comments from the reviewers in a way that is mostly satisfactory. Given that the research question now is stated as "how does someone become a leader in public health in the context of Nepal", I believe that the results and the discussion sections are appropriate treatments of this. The present text is an explorative documentation of how individuals develop their public health leadership meeting resource shortages, caste system and gender inequality. In my opinion, this merits publication. I still think that the authors could have made their case and relevance even a bit stronger by outlining the situation for the Nepalese public health system. There seem to be some assumptions in the text building on tacit knowledge about the state of public health in Nepal, and the challenges facing a developing country with a population that has needs on many levels. This study brings out some of the mechanisms here, such as the sense of value-based engagement and the need to develop informal influential capabilities. The non-Nepalese reader would probably find a short description of the present-day challenges interesting to understand the full impact of the journey these public health leaders are making, and it might improve the future value of the article. However I leave it to the authors whether they would want to make some amendments in this direction or whether they think the present form of the article is sufficient for their aims.

7. PLOS authors have the option to publish the peer review history of their article (what does this mean?). If published, this will include your full peer review and any attached files.

Reviewer #1: No

Reviewer #3: **Yes: **Jan Ketil Arnulf

---

## [Editor Report · Acceptance letter]

28 Oct 2021

PONE-D-21-08622R1 

Leadership development among public health officials in Nepal: A grounded theory 

Dear Dr. Subedi:

I'm pleased to inform you that your manuscript has been deemed suitable for publication in PLOS ONE. Congratulations! Your manuscript is now with our production department. 

Kind regards, 

on behalf of

Professor Sharon Mary Brownie 

Academic Editor

PLOS ONE